# GSK5182, 4-Hydroxytamoxifen Analog, a New Potential Therapeutic Drug for Osteoarthritis

**DOI:** 10.3390/ph13120429

**Published:** 2020-11-27

**Authors:** Yunhui Min, Dahye Kim, Godagama Gamaarachchige Dinesh Suminda, Xiangyu Zhao, Mangeun Kim, Yaping Zhao, Young-Ok Son

**Affiliations:** 1Interdisciplinary Graduate Program in Advanced Convergence Technology and Science, Jeju National University, Jeju City 63243, Korea; reinise4011@jejunu.ac.kr (Y.M.); godagama@jejunu.ac.kr (G.G.D.S.); zhaoxianguy@jejunu.ac.kr (X.Z.); 2Department of Animal Biotechnology, Faculty of Biotechnology, College of Applied Life Sciences, Jeju National University, Jeju City 63243, Korea; dahyekim@jejunu.ac.kr (D.K.); aksxhs123@jejunu.ac.kr (M.K.); 3School of Chemistry and Chemical Engineering, Frontiers Science Center for Transformative Molecules, Shanghai Jiao Tong University, Shanghai 200240, China; ypzhao@sjtu.edu.cn; 4Bio-Health Materials Core-Facility Center, Jeju National University, Jeju City 63243, Korea; 5Practical Translational Research Center, Jeju National University, Jeju City 63243, Korea

**Keywords:** GSK5182, ERRγ, cartilage degeneration, osteoarthritis

## Abstract

Estrogen-related receptors (ERRs) are the first identified orphan nuclear receptors. The ERR family consists of ERRα, ERRβ, and ERRγ, regulating diverse isoform-specific functions. We have reported the importance of ERRγ in osteoarthritis (OA) pathogenesis. However, therapeutic approaches with ERRγ against OA associated with inflammatory mechanisms remain limited. Herein, we examined the therapeutic potential of a small-molecule ERRγ inverse agonist, GSK5182 (4-hydroxytamoxifen analog), in OA, to assess the relationship between ERRγ expression and pro-inflammatory cytokines in mouse articular chondrocyte cultures. ERRγ expression increased following chondrocyte exposure to various pro-inflammatory cytokines, including interleukin (IL)-1β, IL-6, and tumor necrosis factor (TNF)-α. Pro-inflammatory cytokines dose-dependently increased ERRγ protein levels. In mouse articular chondrocytes, adenovirus-mediated ERRγ overexpression upregulated matrix metalloproteinase (MMP)-3 and MMP-13, which participate in cartilage destruction during OA. Adenovirus-mediated ERRγ overexpression in mouse knee joints or ERRγ transgenic mice resulted in OA. In mouse joint tissues, genetic ablation of *Esrrg* obscured experimental OA. These results indicate that ERRγ is involved in OA pathogenesis. In mouse articular chondrocytes, GSK5182 inhibited pro-inflammatory cytokine-induced catabolic factors. Consistent with the in vitro results, GSK5182 significantly reduced cartilage degeneration in ERRγ-overexpressing mice administered intra-articular Ad-*Esrrg*. Overall, the ERRγ inverse agonist GSK5182 represents a promising therapeutic small molecule for OA.

## 1. Introduction

Osteoarthritis (OA) is the most well-known arthritic disease. OA primarily involves chronic inflammation of the articular cartilage [1,2,3] and shows pathological changes in the synovial membrane, meniscus, and infrapatellar fat pad with low-grade inflammation [4,5]. OA research has shifted from being considered a “wear and tear” disease to a “metabolic” disease [6,7]. OA is caused by an imbalance in anabolic and catabolic factors [8]. These processes are involved in risk factors such as mechanical injury, genetic factors, aging, obesity, gender, and metabolic disorders [9,10]. Environmental or genetic OA risk elements change biochemical mechanisms in articular chondrocytes, resulting in loss of the extracellular matrix (ECM). Among catabolic factors, matrix metalloproteinase (MMP)-3, MMP-13, and a disintegrin-like and metallopeptidase with thrombospondin type 1 motif 5 (ADAMTS5) are known to play important roles in cartilage destruction [11,12,13,14]. Catabolic elements of the pro-inflammatory cytokine interleukin (IL)-1β, IL-6, and tumor necrosis factor (TNF)-α activate transcription of nuclear kappa B (NF-κB), and control the loss of the matrix in articular cartilage through upregulation of MMP-3, MMP-13, and ADAMTS5 [3,15,16]. These cytokines also increase intracellular reactive oxygen species (ROS) concentration, thereby inducing chondrocyte apoptosis [17]. Hypoxia-inducible factor (HIF)-2α (encoded by *Epas1*) [18,19] and the ZIP8 (encoded by *Slc39a8*) [20], which are crucially regulated during OA pathogenesis, were upregulated by pro-inflammatory cytokines. Even though pro-inflammatory cytokines are primary therapeutic targets for osteoarthritis, only a few clinical studies have been completed [15,17].

The estrogen-related receptors (ERRs) consist of ERRα, -β, and -γ [21,22]. ERRs are orphan nuclear receptors that possess high similarity sequence DNA-binding domains of estrogen receptors (ERs) [22]. However, ERRs do not bind to 17β-estradiol as an estrogen ligand [23]. ERRs are involved in various metabolic processes, including alcohol, bone, cholesterol, glucose, iron, and lipid metabolism [24]. They are expressed in the liver, muscle, heart, and bone [23,24]. Our group has reported that ERRγ is a novel catabolic regulator of OA pathogenesis [7]. GSK5182 (a 4-hydroxy tamoxifen analog) is a selective inverse agonist of ERRγ [25] and inhibits *Esrrg* transcriptional activity by recruiting small heterodimer partner (SHP)-interacting leucine zipper protein (SMILE) [26]. The inhibitory effects of GSK5182 on pro-inflammatory cytokine-mediated OA pathogenesis are limited [24,27]. Therefore, the aim of this study was to elucidate whether the small molecule GSK5182 is a potential therapeutic molecule for OA pathogenesis.

## 2. Results

### 2.1. ERRγ Is Upregulated in Pro-Inflammatory Cytokine Exposed Chondrocytes

To explore the possible association between ERRγ and inflammatory conditions in OA pathogenesis, various OA-associated inflammatory cytokines, including IL-1β [12,15], IL-6 [28,29], and TNF-α [30,31], were used to treat primary cultured chondrocytes. Reverse transcription-polymerase chain reaction (RT-PCR) analyses revealed that ERRγ was remarkably increased in IL-1β, IL-6, and TNF-α-exposed chondrocytes (Figure 1A–C). Quantification of these pro-inflammatory cytokine mRNA expression levels was performed by qRT-PCR (Figure 1D). The highest expression time was 0.5 h for IL-1β, 1 h for IL-6, and 2 h for TNF-α. After reaching a peak, ERRγ expression was reduced. Notably, mRNA and protein levels of matrix degradation enzymes, such as MMP-3 and MMP-13, and ERRγ increased with increasing IL-1β concentrations (Figure 1E,F). 

### 2.2. Pro-Inflammatory Cytokines Increased ERRγ Protein, and ERRγ Overexpression Caused Upregulation of MMP Expression in Articular Chondrocytes

To examine whether pro-inflammatory cytokines upregulate ERRγ protein levels, the chondrocytes were exposed to the indicated concentrations of IL-1β, IL-6, and TNF-α for 24 h (Figure 2A–C). ERRγ expression levels induced by pro-inflammatory cytokines were significantly increased in a dose-dependent manner. In chondrocytes overexpressing ERRγ via infection with Ad-*Esrrg*, MMP-3 and MMP-13 mRNA levels were dramatically elevated without affecting MMP-12 and ADAMTS5 (Figure 2D). The extracellular protein levels of MMP-3 and MMP-13 increased following dose- and time-dependent ERRγ overexpression (Figure 2E,F). Collectively, pro-inflammatory cytokines induced ERRγ expression, and the overexpression of ERRγ caused MMP-3 and MMP-13 expression at both the mRNA and protein levels.

### 2.3. The Ectopic Expression or Genetic Ablation of ERRγ in the Mice

To investigate the role of ERRγ in OA pathogenesis in vivo, we ectopically overexpressed ERRγ in the knee joint tissues of 12-week-old male mice. We employed an intra-articular (IA) injection to deliver an adenovirus expressing *ERRγ* (*Ad-Esrrg*). The adenovirus delivery system to joint tissues using IA injection has been well established [6,7,32]. ERRγ overexpression was induced by IA injection (three weekly IA injections) of Ad-*Esrrg*, which induced a loss of glycosaminoglycans in articular cartilage above the tidemark, as determined by safranin O staining (Figure 3A, left panel). Cartilage degeneration was quantified using the Osteoarthritis Research Society International (OARSI) grade. ERRγ overexpression significantly increased the OARSI grade (*p* < 0.0001) (Figure 3A, right panel). To further examine ERRγ cartilage-specific functions in OA pathogenesis, we used cartilage-specific ERRγ Tg mice (Col2a1-*Esrrg*) [7]. Compared with wildtype (WT) littermates, destabilization of the medial meniscus (DMM)-operated Col2a1-*Esrrg* Tg mice exhibited dramatically more cartilage damage, as shown by safranin O staining and the OARSI grade (*p* < 0.0001) (Figure 3B). Other symptoms of OA, including subchondral sclerosis and osteophyte formation, were also dramatically enhanced in Col2a1-*Esrrg* Tg mice when compared with WT littermates (Figure 3B). Collectively, the results of our current experiments demonstrated that ERRγ is a key player in OA pathogenesis. Additionally, we used ERRγ-knockout (KO) mice as a reverse approach. ERRγ-null mice demonstrate embryonic lethality [33], therefore we used heterozygous mice (*Esrrg*^+/−^) for the OA experiments. ERRγ-knockout (KO) mice revealed that DMM-induced cartilage erosion, osteophyte formation, and subchondral bone sclerosis were dramatically attenuated in *Esrrg*^+/−^ mice (*p* < 0.0001) (Figure 3C). This result supported the conclusion that ERRγ is an important catabolic regulator in OA pathogenesis.

### 2.4. Inhibition of ERRγ by GSK5182 Attenuates Experimental OA Pathogenesis

Finally, we investigated whether GSK5182 could be a possible therapeutic molecule against OA. Pro-inflammatory cytokines such as IL-1β, IL-6, and TNF-α upregulated ERRγ expression as well as the expression levels of MMP-3 and MMP-13 in chondrocytes, therefore we evaluated treatment with GSK5182, an inverse agonist of ERRγ [34]. We observed that treatment with GSK5182 significantly inhibited IL-1β, IL-6-, or TNF-α induced upregulation of ERRγ, MMP-3, and MMP-13 in primary cultured chondrocytes (Figure 4A–C). Furthermore, GSK5182 treatment inhibited the expression of ERRγ, MMP-3, and MMP-13 in *Ad-Esrrg*-transfected chondrocytes (Figure 4D). To confirm that ERRγ is a potential therapeutic target molecule against OA, we delivered Ad-*Esrrg* with GSK5182 by IA injection to mice knee joint tissues. We observed that GSK5182 significantly reduced ERRγ overexpression-mediated cartilage destruction (*p* < 0.05) (Figure 4E) concomitant with ERRγ expression in the knee joints (Figure 4F). Collectively, these results suggested the possibility of utilizing GSK5182 as a therapeutic OA drug.

## 3. Discussion

OA is the most well-known form of arthritis; its symptoms include cartilage destruction, synovial inflammation, osteophyte formation, and subchondral bone sclerosis [5,8,35]. Moreover, OA is an arthropathy and a leading cause of pain and disability with a sizable socioeconomic cost. However, no effective therapies for OA have been developed. Our group has reported that ERRγ acts as a catabolic regulator of cartilage degeneration and OA pathogenesis [7]. In addition, we have demonstrated that the inverse agonist of ERRγ, GSK5182, inhibits OA pathogenesis in a mouse model [7]. However, previous studies have reported limited information regarding the relationship between pro-inflammatory cytokines, ERRγ expression, and GSK5182. Therefore, we further investigated the relationship between pro-inflammatory cytokines and ERRγ expression, and GSK5182 function in the pro-inflammatory cytokine-mediated cartilage catabolism in the OA joint. 

Previously, we failed to define whether pro-inflammatory cytokines induced ERRγ expression [7]. IL-1β, TNF-α, and IL-6 appear to be the central pro-inflammatory cytokines involved in OA pathophysiology [15,28,29]. It has been reported that IL-1β and TNF-α are elevated in the synovial fluid and synovial membrane, subchondral bone, and cartilage during OA [3,15,16]. These cytokines suppress type II collagen, proteoglycan, and aggrecan expression while stimulating MMP-1, MMP-3, and MMP-13 expression [17,36,37,38,39,40]. Additionally, IL-6 levels are highly elevated in the synovial fluid and serum of patients with OA, with MMP functions [41]. The current study demonstrated that the mRNA and protein ERRγ levels were significantly increased in chondrocytes exposed to IL-1β, IL-6, and TNF-α (Figure 1 and Figure 2). Notably, OA pathogenesis is mediated by an imbalance between anabolic and catabolic factors. OA-causing primary mechanisms include mechanical stresses (joint instability and injury), which induce the activation of biochemical pathways in chondrocytes, resulting in a loss of ECM by matrix metalloproteinases (MMPs) and aggrecanases (ADAMTSs). MMP-3, MMP-13, and ADAMTS5 are known to play crucial roles in OA cartilage destruction [12,13,14]. Our study showed that ERRγ overexpression via transduction with Ad-*Esrrg* dramatically elevated the mRNA or protein levels of MMP-3 and MMP-13 (Figure 2D–F). These results suggest that in articular chondrocytes, pro-inflammatory cytokines, ERRγ, and MMPs are closely associated. Based on our previous report and other available evidence, ERRγ might directly regulate MMP transcription [7,42].

We further analyzed the genetic function of ERRγ in vivo in OA pathogenesis systems. ERRγ (NR3B3, *Esrrg*) is one of the ERR isoforms (ERRα; NR3B1, *Esrra*), (ERRβ; NR3B2, *Esrrb*), which was first identified as an orphan nuclear receptor [21]. ERRs are closely related to the ER without binding to the ER ligand but share high homology in their DNA-binding domain [21]. ERRα positively regulates osteoblast differentiation and bone formation [21], but ERRγ has demonstrated the opposite function [33,43]. ERRs have functions in chondrocytes and OA. For example, ERRα plays dual roles in OA chondrocytes; ERRα increases pro-chondrogenic factor (SOX9) and cartilage-degenerative factor (MMP-13) in response to pro-inflammatory factors [7,42,44]. Our animal studies demonstrated that ERRγ overexpression by either adenovirus delivery system (Ad-*Esrrg*) or cartilage-specific ERRγ Tg mice (Col2a1-*Esrrg*) enhanced cartilage degeneration, osteophyte formation, and subchondral bone sclerosis, which are hallmarks of OA pathogenesis (Figure 3A,B). In contrast, an opposite phenomenon was observed in ERRγ-knockout (KO) mice (Figure 3C). This result indicated that ERRγ is a crucial mediator in OA pathogenesis. Finally, we focused on whether the ERRγ inverse agonist GSK5182 blocks pro-inflammation mediated MMP-3 and MMP-13 expression as well as ERRγ expression. The ERRγ binding activity for synthetic ligands diethylstilbestrol (DES), tamoxifen (TAM), and 4-hydroxytamoxifen (4-OHT) was evaluated and 4-OHT was shown to be the most specific and had high binding affinity with ERRγ at the micro-molar level [45]. Synthetic GSK5182, a 4-OHT analog, was developed to exhibit higher affinity (IC50 = 79 nM) for ERRγ [46] and was shown to regulate the transcriptional activity of ERRγ [47,48]. As expected, treatment with GSK5182 dramatically inhibited IL-1β, IL-6-, or TNF-α induced MMP-3 and MMP-13 expression in primary cultured chondrocytes (Figure 4A–C). Notably, GSK5182 unquestionably inhibited ERRγ overexpression-mediated MMP-3 and MMP-13 expression as well as ERRγ expression (Figure 4D). It could be postulated that GSK5182 regulates transcription of ERRγ as well as post-transcriptional regulation [47,48,49]. Following GSK5182 delivery to the mouse knee joint, protection against ERRγ-induced cartilage degeneration was observed (Figure 4E). Treatment with tamoxifen induced cancellous bone and longitudinal growth, while treatment with GSK5182 reduced DMM-induced bone remodeling including cartilage destruction, osteophyte development, and subchondral bone sclerosis [7,50]. Data from cellular and animal studies revealed that GSK5182 is a potential therapeutic drug for OA by blocking inflammatory pathways. 

## 4. Materials and Methods 

### 4.1. Chemicals and Laboratory Ware 

Unless specified otherwise, all chemicals and laboratory wares were purchased from Sigma Chemical Co., (St. Louis, MO, USA) and Falcon Labware (Becton-Dickinson, Franklin Lakes, NJ, USA), respectively. Dulbecco’s modified Eagle’s medium (DMEM) and fetal bovine serum (FBS) were purchased from Gibco Co., (Gibco BRL, New York, NY, USA). 

### 4.2. Experimental OA in Mice 

C57BL/6J (18 mice), ERRγ total-KO (B6.129P2-*Esrrg*^tm1Dgen^/Mmnc; MMRRC, Davis, CA, USA) (16 mice), and cartilage-specific transgenic mice for ERRγ (25 mice) were used for experimental OA investigations [7]. The cartilage-specific ERRγ Tg mice (Col2a1-*Esrrg*) were generated using the Col2a1 enhancer and promoter [7]. All experiments were approved by the Jeju National University Animal Care and Use Committee (2020-0001). To avoid any developmental effects resulting from hormonal differences, the OA experiments were completed using 12-week-old male mice. OA was induced by DMM surgery [28,51,52] or by an IA injection (once weekly for three weeks) of adenovirus (1 × 10^9^ plaque-forming units (PFUs) in a total volume of 10 μL) expressing ERRγ (Ad-*Esrrg*) [7,18,20]. Mouse knee joints were harvested 8 weeks after DMM, and 3 or 8 weeks after the first IA injection for histological and biochemical analyses. 

### 4.3. Primary Culture of Articular Chondrocytes, Adenoviruses, Infection of Chondrocytes, and IA Injection 

Chondrocytes were isolated from femoral condyles and tibial plateaus of 4-day-old mice (n = 12) by digesting cartilage tissue with 0.2% collagenase (Sigma, Darmstadt, Germany) [7,18,28,51,52,53]. The passage “0” primary chondrocytes (3 × 10^5^/30 mm culture dish) were maintained as a monolayer in DMEM (Gibco, Waltham, MA, USA) supplemented with 10% FBS and antibiotics (penicillin G and streptomycin; Gibco, Waltham, MA, USA). The adenovirus expressing mouse ERRγ (Ad-*Esrrg*) was kindly provided by Dr. Choi (Chonnam National University, Gwangju, South Korea) [26,34]. Mouse articular chondrocytes (3 × 10^5^) were cultured for two days, infected with various concentrations of Ad-*Esrrg* adenoviruses [200–800 MOI (multiplicity of infection)] for 2 h, and cultured in the presence or absence of GSK5182 (2.5. 10 μM) for an additional 24 h. Cells were treated with various inflammatory cytokines (IL-1β; 0.1–1 ng/mL, IL-6; 10–50 ng/mL, and TNF-α; 10–50 ng/mL) for 24 h. Adenovirus (1 × 10^9^ PFUs in a total volume of 10 μL) was injected into the knee joints of mice once per week for 3 weeks. Mice were sacrificed 3 weeks after the first adenovirus injection. 

### 4.4. Reverse Transcription-Polymerase Chain Reaction (RT-PCR) 

Total RNA was extracted from primary cultured chondrocytes using TRI reagent (Molecular Research Center Inc., Cincinnati, OH, USA). The quality and concentration of RNA were evaluated using a NanoDrop™ 2000 Spectrophotometer (Thermo Scientific, Waltham, MA, USA). The RNA was reverse transcribed, and the resulting cDNA was amplified by PCR or CFX96™ Real-Time System (BIO-RAD, Hercules, CA, USA,in Bio-Health Materials Core-Facility, Jeju National University) using SYBR premix ExTaq reagents (TaKaRa Bio, Mountain View, CA, USA). The PCR primers and experimental conditions are summarized in Table 1. Glyceraldehyde-3-phosphate dehydrogenase (GAPDH) was used as internal control. 

### 4.5. Western Blotting

Total cell lysates were prepared in lysis buffer [150 mM NaCl, 1% NP-40, 50 mM Tris, 0.2% sodium dodecyl sulfate (SDS), 5 mM NaF] and used to detect ERRγ. Secreted proteins (MMP-3 and MMP-13) were detected after precipitation with trichloroacetic acid (TCA) from 900 μL of serum-free conditioned medium. All lysis buffers contained a cocktail of protease inhibitors and phosphatase inhibitors (Roche, Basel, Switzerland). Target bands were quantified using ImageJ densitometry software (NIH, Bethesda, MD, USA). The following antibodies were used for Western blotting: rabbit polyclonal anti-ERRγ (1:200 dilution; sc-66883) from Santa Cruz Biotechnology (Dallas, TX, USA), anti-MMP-3 (clone EP1186Y, 1 μg mL^−1^, ab52915), and anti-MMP-13 (clone EP1263Y, 1:1000 dilution; ab51072) from Abcam Plc. (Cambridge, MA, USA).

### 4.6. Histology

Mouse knee joints presenting with experimental OA were fixed in 4% paraformaldehyde, decalcified in 0.5 M EDTA, and embedded in paraffin. The paraffin blocks were sectioned at a thickness of 5 μm, and sections were deparaffinized in xylene, hydrated with graded ethanol, and stained with safranin O. Cartilage destruction was scored by five observers under blinded conditions using the OARSI scoring system (grades 0–6) [7,18,20,54] The results of OARSI grade scoring represent the mean of the maximum score in each mouse, and the representative safranin O-stained image was selected from the most advanced lesion among serial sections.

### 4.7. Statistical Analysis

All statistical analyses were performed using IBM SPSS Statistics 21 (IBM Corp., Armonk, NY, USA). Data from the cell-based in vitro assays were evaluated using two-tailed Student’s *t*-tests with unequal sample sizes and variances and two-way analysis of variance (ANOVA) with post-hoc tests (LSD) for pairwise comparisons and multi-comparisons, respectively. Data collected from the mouse experiments were analyzed using the non-parametric Mann–Whitney U test. Data distribution was evaluated for normalcy using the Shapiro–Wilk test. Herein, “n” indicates the number of independent experiments or mice. Significance was accepted at the 0.05 level of probability (*p* < 0.05).

## Figures and Tables

**Figure 1 pharmaceuticals-13-00429-f001:**
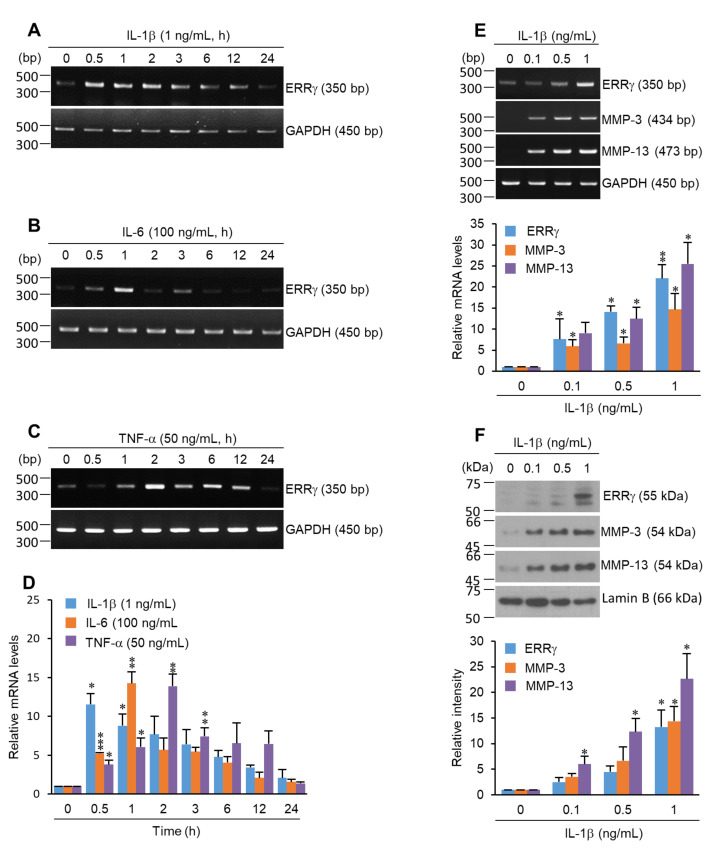
Pro-inflammatory cytokines induce ERRγ expression. The primary cultured articular chondrocytes were exposed to IL-1β (1 ng/mL) (**A**), IL-6 (100 ng/mL) (**B**), and TNF-α (50 ng/mL) (**C**) for the indicated time course, and ERRγ expression was analyzed using RT-PCR and quantified by qRT-PCR (**D**). Chondrocytes treated with IL-1β (0–1 ng/mL), and the expression levels of ERRγ, MMP-3, and MMP-13 were analyzed with RT-PCR and quantified by qRT-PCR (**E**). Protein levels were analyzed by Western blotting with semi-quantification (**F**). The results are representative of three independent experiments from different pups. Values are presented as mean ± standard error of the mean (SEM) (* *p* < 0.05, ** *p* < 0.01, and *** *p* < 0.001). One-way ANOVA was performed. GAPDH and Lamin B were used as internal markers. ERRγ, estrogen-related receptor γ; IL-1β, interleukin-1β, IL-6, interleukin 6; TNF-α, tumor necrosis factor-α; MMP-3, matrix metalloproteinase-3; MMP-13, matrix metalloproteinase-13; RT-PCR, reverse transcription-polymerase chain reaction; GAPDH, glyceraldehyde-3-phosphate dehydrogenase.

**Figure 2 pharmaceuticals-13-00429-f002:**
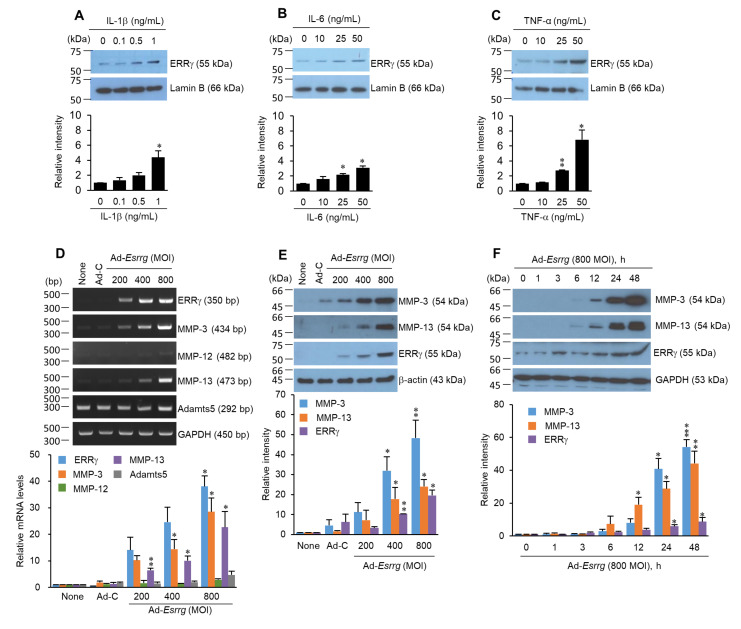
ERRγ is an inducer of MMP-3 and MMP-13 in articular chondrocytes. Primary cultured articular chondrocytes were exposed to IL-1β (0–1 ng/mL) (**A**), IL-6 (0–50 ng/mL) (**B**), and TNF-α (0–50 ng/mL) (**C**) for 24 h. ERRγ protein levels were quantified by Western blotting. (**D**) mRNA levels of ERRγ and catabolic factors (MMP-3, MMP-12, MMP-13, and ADAMTS5) were analyzed by RT-PCR and qRT-PCR in primary cultured chondrocytes infected with Ad-*C* (800 MOI) or the indicated MOI of Ad-*Esrrg* for 36 h. (**E**) The protein levels of ERRγ, MMP-3, and MMP-13 were analyzed by Western blot with semi-quantification in primary cultured chondrocytes infected with Ad-*C* (800 MOI) or the indicated MOI of Ad-*Esrrg* for 36 h. (**F**) The protein levels of ERRγ, MMP-3, and MMP-13 were analyzed by Western blotting with semi-quantification in primary cultured chondrocytes infected with Ad-*Esrrg* (800 MOI) for the indicated hours. The results are representative of three independent experiments from different pups. Values are presented as mean ± SEM (* *p* < 0.05, ** *p* < 0.01, and *** *p* < 0.001). One-way ANOVA. GAPDH, β-actin, and Lamin B were used as internal markers. ERRγ, estrogen-related receptor γ; IL-1β, interleukin-1β, IL-6, interleukin 6; TNF-α, tumor necrosis factor-α; MOI, multiplicity of infection; MMP-3, matrix metalloproteinase-3; MMP-13, matrix metalloproteinase-13.

**Figure 3 pharmaceuticals-13-00429-f003:**
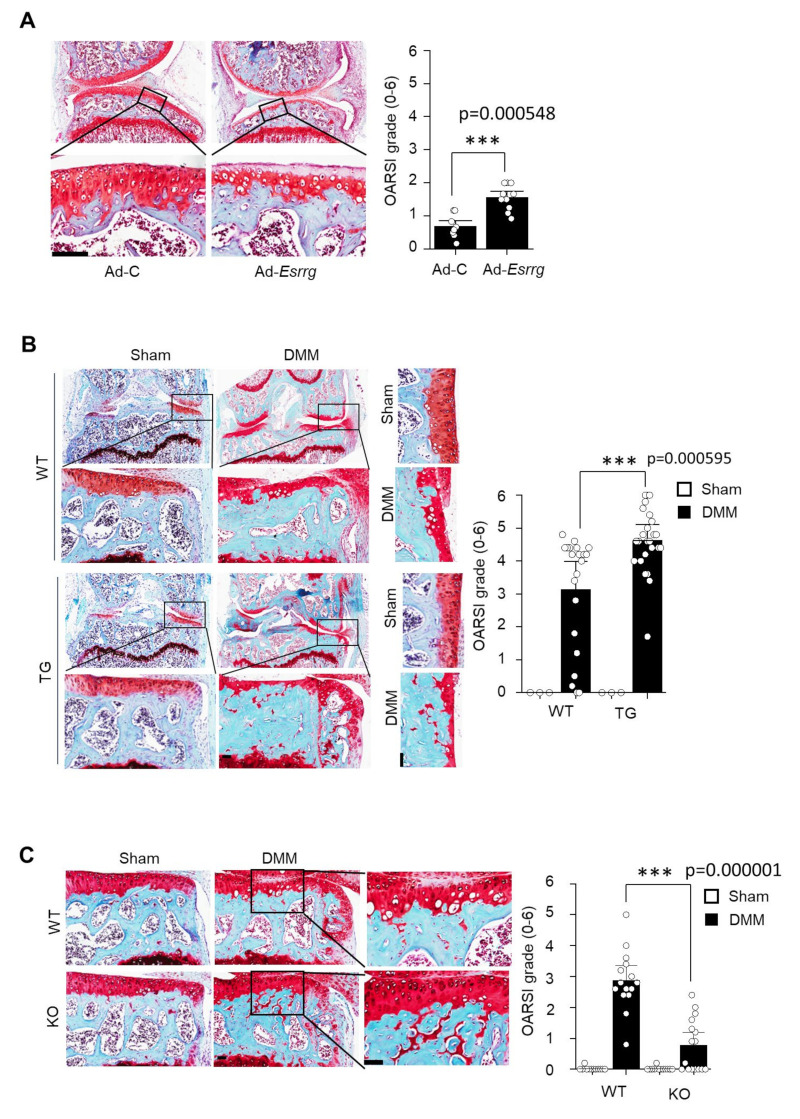
ERRγ regulates OA pathogenesis (**A**) ERRγ overexpression causes cartilage degeneration in the joint tissue of mice. C57BL/6 mice were injected intraarticularly with Ad-C (control; n = 8) or Ad-*Esrrg* (n = 10) in the knee joint. Three weeks after the first injection, lateral sections were then created and changes in the cartilage area were analyzed by safranin O staining. Values are presented as the mean ± SEM and have been evaluated using the Mann–Whitney U test (*U* = 4.0, *p* = 0.000548, *r* = −1.0150). (**B**) ERRγ Tg (Col2a1-*Esrrg*) mice exhibit an enhanced OA phenotype. ERRγ Tg (Col2a1-*Esrrg*) mice and WT littermates underwent DMM surgery. Frontal sectioning was performed and OARSI grade was quantified in WT (n = 21) and Col2a1-*Esrrg* TG (n = 25) mice 8 weeks after sham operation or DMM surgery. Representative images of safranin O-stained joint sections showing the whole joint (40×), subchondral bone sclerosis, osteophyte size (200×), and cartilage (400×). Values are presented as the mean ± SEM and have been evaluated using the Mann–Whitney U test (*U* = 107.5, *p* = 0.000595, *r* = −0.6868). (**C**) Genetic knockdown of *Esrrg* attenuates OA pathogenesis in mice. ERRγ knockdown (*Esrrg^+/−^*) mice and WT littermates underwent DMM surgery. Frontal sections were created and OARSI grade was scored in WT (n = 15) and *Esrrg^+/−^* (n = 16) mice. Values are presented as the mean ± SEM and have been evaluated using the Mann–Whitney U test (*U* = 11.5, *p* = 0.000001, *r* = −1.0795). *** *p* < 0.0001. Two-tailed *t*-test and Mann–Whitney U test. Scale bar: 50 μm. OA, osteoarthritis; ERRγ, estrogen-related receptor γ; DMM, destabilization of the medial meniscus; WT, wildtype; OARSI, Osteoarthritis Research Society International.

**Figure 4 pharmaceuticals-13-00429-f004:**
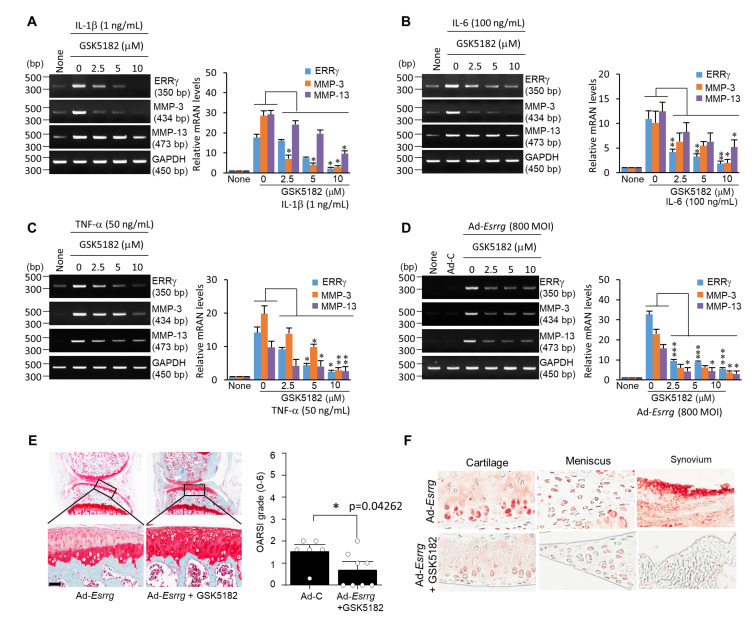
GSK5182, an inverse agonist ERRγ, suppresses OA pathogenesis in vitro and in vivo. RT-PCR or qRT-PCR analyses of ERRγ, MMP-3, and MMP-13 expression in chondrocytes treated with IL-1β (1 ng mL^−1^) (**A**), IL-6 (100 ng mL^−1^) (**B**), and TNF-α (50 ng mL^−1^) (**C**) at indicated concentrations of GSK5182. (**D**) mRNA levels of ERRγ and MMPs (MMP-3 and MMP-13) were analyzed by RT-PCR or qRT-PCR in primary cultured chondrocytes infected with Ad-*Esrrg* (800 MOI) and combined with indicated concentrations of GSK5182 for 36 h. The results are representative of three independent experiments from different pups. Values are presented as mean ± standard error of the mean (SEM) (* *p* < 0.05, ** *p* < 0.01, and *** *p* < 0.001 vs. only cytokine treatment). One-way ANOVA was performed. GAPDH was used as an internal marker. (**E**) The mice were administered an IA injection of Ad-*Esrrg* (n = 6) to overexpress ERRγ in the joint tissues. Mice injected with Ad-*Esrrg* were administered an IA injection of GSK5182 (n = 8) and sacrificed 3 weeks after the IA injection. Lateral sectioning was performed and the OARSI grade was calculated after safranin O staining. (**F**) Expression of ERRγ was evaluated by IHC, and the values are presented as the mean ± SEM and have been evaluated using the Mann–Whitney U test (*U* = 8.0, *p* = 0.04262, *r* = −7.4599). * *p* < 0.05. Scale bar: 50 μm. ERRγ, estrogen-related receptor γ; OA, osteoarthritis; IL-1β, interleukin-1β, IL-6, interleukin 6; TNF-α, tumor necrosis factor-α; IA, intra-articular; MOI, multiplicity of infection; MMP-3, matrix metalloproteinase-3; MMP-13, matrix metalloproteinase-13; RT-PCR, reverse transcription-polymerase chain reaction.

**Table 1 pharmaceuticals-13-00429-t001:** PCR primers and conditions.

Genes	Strand	Primer Sequences	Size (bp)	AT (°C)	Origin
*ADAMTS5*	S AS	5′-GCCATTGTAATAACCCTGCACC-3′ 5′-TCAGTCCCATCCGTAACCTTTG-3′	292	58	Mouse
*Esrrg*	S AS	5′-AAGATCGACACATTGATTCCAGC-3′ 5′-GCTTCACATGATGCAACCCC-3′	350	64	Mouse
*GAPDH*	S AS	5′-TCACTGCCACCCAGAAGAC-3′ 5′-TGTAGGCCATGAGGTCCAC-3′	450	58	Mouse
*MMP-3*	S AS	5′-AGGGATGATGATGCTGGTATGG-3′ 5′-CCATGTTCTCCAACTGCAAAGG-3′	434	58	Mouse
*MMP-12*	S AS	5′-CCCAGAGGTCAAGATGGATG-3′ 5′-GGCTCCATAGAGGGACTGAA-3′	482	60	Mouse
*MMP-13*	S AS	5′-TGATGGACCTTCTGGTCTTCTGG-3′ 5′-CATCCACATGGTTGGGAAGTTCT-3′	473	58	Mouse

AT, annealing temperature; S, sense; AS, antisense.

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
