# Peer review of "GSK5182, 4-Hydroxytamoxifen Analog, a New Potential Therapeutic Drug for Osteoarthritis"

_pharmaceuticals, 2020, doi:10.3390/ph13120429_

Round 1

Reviewer 1 Report

In the manuscript entitled “GSK5182, 4-hydroxytamofifen analog, is a new potential therapeutic drug for osteoarthritis” by Min et al, the authors have used in vitro and in vivo approaches to study the effect of ERRg overexpression, depletion and effect of inhibitor. The study is very interesting and appears to be follow up study to their previous published work (PMID: 29247173). See my comments below:

Abstract: line 36, “Inconsistent” or “Consistent”

Highlight the advances made in this study compared to the previous study by the same author.

Introduction: Cite most recent references on the role of inflammation in OA.

Results.

Authors should provide qPCR data for gene expression experiments as the amplicon size of the RT-PCR is very high which is not suitable for quantification.

Methods: Explain the statistical analyses in detail. Why the authors used Mann-Whitney?

Author Response

Reviewer #1:

In the manuscript entitled "GSK5182, 4-hydroxy tamoxifen analog, is a new potential therapeutic drug for osteoarthritis" by Min et al, the authors have used in vitro and in vivo approaches to study the effect of ERRg overexpression, depletion and effect of inhibitor. The study is very interesting and appears to be follow up study to their previous published work (PMID: 29247173). See my comments below:

Comment 1.

Abstract: line 36, “Inconsistent” or “Consistent”

[Response]

It is a typographic error. We revised “Inconsistent” to “Consistent”.

Comment 2.

Highlight the advances made in this study compared to the previous study by the same author.

[Response]

N/A

Comment 3.

Introduction: Cite most recent references on the role of inflammation in OA.

[Response]

According to the reviewer’s comment, we cited recent references in the Introduction section (line 45, 48, 58, 68)

Results.

Comment 4.

Authors should provide qPCR data for gene expression experiments as the amplicon size of the RT-PCR is very high which is not suitable for quantification.

[Response]

According to the reviewer’s comment, we provided all qPCR data for gene expression.

Comment 5.

Methods: Explain the statistical analyses in detail. Why the authors used Mann-Whitney?

[Response]

The Mann-Whitney U test is used to compare differences between two independent groups when the dependent variable is either ordinal or continuous but not normally distributed. The Mann-Whitney U test is often considered the nonparametric alternative to the independent t-test, although this is not always the case. The Mann-Whitney U test is a commonly used statistical analysis of OA parameters such as OARSI grade, osteophyte maturity, ICR grade, Mankin score, synovial inflammation, a clinical score of CIA model, pannus formation, mouse pain behavior (von frey test, hot plate assay), etc.

Reviewer 2 Report

The aim of the study should be added at the end of the introduction. The authors should explain clearly what was demonstrated in their previous paper and what is the novelty that they want to investigate in this manuscript because the two papers are very similar in my opinion. In the previous paper, the authors reported more information on the mice experiments (for example osteophyte size and subchondral bone thickness). It seems that the focus should be the use of GSK5182 in this manuscript. However, this inhibitor was used also in the previous paper (reference 4) where the authors reported also the effect of the inhibitor in DMM model, which is absent here.

The introduction on osteoarthritis should be improved. For example they report that “OA is primarily chronic inflammation  in the articular cartilage [1,2].”. However, OA is a whole joint disease involving all joint tissue such as synovial membrane, infrapatellar fat pad (not only articular cartilage) and it is characterized by low grade inflammation. The authors should also cite appropriate references.

Line 47,: the authors should add appropriate references.

Lines 55-56: it is questionable this sentence. No references have been provided.

Lines 57-59: the sentences should be rewritten.

Line 61: the font is different.

The methods are short and do not provide all information. The paragraph 4.2 should be improved. The authors should report the number of mice used for each experiment, what kind of tissues were collected etc. Paragraph 4.3: the authors should report the number of cells seeded for the treatments, the plates used and the concentrations of the treatments. Authors should specify the number of mice used to isolate the chondrocytes and the months of mice.

Did the authors used 2ˆ(–delta delta CT) to elaborate real time PCR data?

Figure 1: could the authors report also graphs illustrating ERRγ expression with appropriate statistics?

It is not clear to me why the authors treated chondrocytes with 1 ng/ml of IL-1beta. The typical concentration used is 10ng/ml. Could the authors justify the concentrations of cytokines used?

Figure 1 E: why there are several bands for ERRγ and MMP3?

All the western blot bands should be quantified with an appropriate software (such as ImageJ) and a graphs should be reported with appropriate statistical analysis. It is not clear why the authors used different housekeeping (GADPH, beta actin and laminin). Could the authors report all the western blot performed as supplementary materials?

In the legend of Figure 3, the authors wrote that WT mice were intraarticularly injected with Ad-C. Could the authors specify what kind of injection was performed in the controls?

The authors cited first figure 3C and then figure 3B. Could the author reorganize the text or the figures in order to cite first B and then C?

Could the authors define: cartilage-specific ERRγ Tg mice (Col2a1-Esrrg)? The authors should give all the information to the reader in order to understand and follow the manuscript. I realize that the authors generated a  cartilage-specific ERRγ Tg mice (Col2a1-Esrrg) using the Col2a1 enhancer and promoter by reading reference number 4.

Figure 3C: OARSI grade was quantified in WT (n=21) and Col2a1-Esrrg TG (n=25) mice 8 weeks after sham operation or DMM surgery. The authors reported the total number of mice for each group (WT (n=21) and Col2a1-Esrrg TG (n=25)). It is not clear how many mice were included in the group “sham operation” and how many in “DMM surgery” for WT and Col2a1-Esrrg TG.

Line 249, the author wrote: we investigated whether ERRγ could be a possible therapeutic molecule against OA. Did the author mean that they investigated whether ERRγ could be a possible therapeutic target (and not molecule) to counteract/slow OA?

Author Response

Reviewer #2:

Comment 1.

The aim of the study should be added at the end of the introduction.

[Response]

Following the reviewer's comment, we added the study's aim at the end of the introduction (lines 67~69).

Comment 2.

The authors should explain clearly what was demonstrated in their previous paper and what is the novelty that they want to investigate in this manuscript because the two papers are very similar in my opinion. In the previous paper, the authors reported more information on the mice experiments (for example osteophyte size and subchondral bone thickness). It seems that the focus should be the use of GSK5182 in this manuscript. However, this inhibitor was used also in the previous paper (reference 4) where the authors reported also the effect of the inhibitor in DMM model, which is absent here.

[Response]

We explained the differences and novelty in this manuscript compared with our previous paper (lines 192-198)

Comment 3.

The introduction on osteoarthritis should be improved. For example, they report that “OA is primarily chronic inflammation in the articular cartilage [1,2].”. However, OA is a whole joint disease involving all joint tissue such as synovial membrane, infrapatellar fat pad (not only articular cartilage) and it is characterized by low grade inflammation. The authors should also cite appropriate references.

[Response]

According to the reviewer’s comment, we have added other OA phenotype with appropriate references (lines 44~45)

Comment 4.

Line 47,: the authors should add appropriate references.

[Response]

According to the reviewer’s comment, we have added appropriate references (lines 47~48)

Comment 5.

Lines 55-56: it is questionable this sentence. No references have been provided.

[Response]

We have clarified this sentence with appropriate references (lines 57~58)

Comment 6.

Lines 57-59: the sentences should be rewritten.

[Response]

According to the reviewer’s comment, we have revised these sentences (lines 59~60)

Comment 7.

Line 61: the font is different.

[Response]

We fixed the font.

Comment 8.

The methods are short and do not provide all information. The paragraph 4.2 should be improved. The authors should report the number of mice used for each experiment, what kind of tissues were collected etc.

[Response]

According to the reviewer’s comment, we have added the information of the experiment, including mice number and tissue information either “Materials and Methods section 4.2” (lines 246~247) or “Figure 3 legends”.

Comment 9.

Paragraph 4.3: the authors should report the number of cells seeded for the treatments, the plates used and the concentrations of the treatments. Authors should specify the number of mice used to isolate the chondrocytes and the months of mice.

[Response]

According to the reviewer’s comment, we have added the information about cells number, plates, and concentrations of the treatment, including the mice number in the “Materials and Methods section 4.3” (lines 256~267)

Comment 10.

Did the authors used 2ˆ(–delta delta CT) to elaborate real time PCR data?

[Response]

According to the reviewer’s comment, we have provided all qPCR data using 2ˆ(–delta delta CT) for gene expression (Figure 1, 2, 4).

Comment 11.

Figure 1: could the authors report also graphs illustrating ERRγ expression with appropriate statistics?

[Response]

According to the reviewer’s comment, we have provided all graphs illustrating ERRγ expression with appropriate statistics (Figure 1).

Comment 12.

It is not clear to me why the authors treated chondrocytes with 1 ng/ml of IL-1beta. The typical concentration used is 10ng/ml. Could the authors justify the concentrations of cytokines used?

[Response]

We used 1 ng/ml of IL-1beta in these experiments. 1 ng/ml concentration of IL-1b is enough to induce catabolic factors (MMP3, MMP13, Adamts5, etc.). Please see the below references.

Ann Rheum Dis. 2017 Feb;76(2):427-434

Nat Commun. 2017 Dec 15;8(1):2133

Nature 2019 Feb;566(7743):254-258

Ann Rheum Dis . 2016 Nov;75(11):2045-2052

Comment 13.

Figure 1 E: why there are several bands for ERRγ and MMP3?

[Response]

Some non-specific bands were detected in ERRγ; however, the target band was real ERRγ; see positive vector control on supplementary figure 1E. In the case of MMP3, we have replaced with a preferred image (Figure 1F).   

Comment 14.

All the western blot bands should be quantified with an appropriate software (such as ImageJ) and a graphs should be reported with appropriate statistical analysis.

[Response]

According to the reviewer’s comments, we have quantified the western blot band with statistical analysis (Fig. 1F, Fig 2A, B, C, E, F).

Comment 15.

It is not clear why the authors used different housekeeping (GADPH, beta actin and laminin).

[Response]

There is no meaning used for different housekeeping proteins. The other investigators used them to prefer housekeeping protein.

Comment 16.

Could the authors report all the western blot performed as supplementary materials?

[Response]

According to the reviewer’s comment, we have demonstrated all the western blot images in the supplementary materials.

Comment 17.

In the legend of Figure 3, the authors wrote that WT mice were intraarticularly injected with Ad-C. Could the authors specify what kind of injection was performed in the controls?

[Response]

We have specified the WT mice to the C57BL/6 mice (line 146). We have intraarticularly injected control adenovirus (Ad-C) as a control. Also, we have added the IA injection method in the “Materials and Methods 4.3 section”.

Comment 18.

The authors cited first figure 3C and then figure 3B. Could the author reorganize the text or the figures in order to cite first B and then C?

[Response]

Following the reviewer’s comment, we have reorganized the figures according to the text (Figure 3).

Comment 19.

Could the authors define: cartilage-specific ERRγ Tg mice (Col2a1-Esrrg)? The authors should give all the information to the reader in order to understand and follow the manuscript. I realize that the authors generated a cartilage-specific ERRγ Tg mice (Col2a1-Esrrg) using the Col2a1 enhancer and promoter by reading reference number 4.

[Response]

Following the reviewer’s comment, we have defined the cartilage-specific ERRγ Tg mice in the “Materials and Methods section 4.2” (lines 248~249).

Comment 20.

Figure 3C: OARSI grade was quantified in WT (n=21) and Col2a1-Esrrg TG (n=25) mice 8 weeks after sham operation or DMM surgery. The authors reported the total number of mice for each group (WT (n=21) and Col2a1-Esrrg TG (n=25)). It is not clear how many mice were included in the group “sham operation” and how many in “DMM surgery” for WT and Col2a1-Esrrg TG.

[Response]

We performed sham operation in the left legs, DMM surgery in the right legs. It is a standard way for DMM surgery. Therefore, the mice number of sham operation or DMM surgery was an equal number.

Comment 21.

Line 249, the author wrote: we investigated whether ERRγ could be a possible therapeutic molecule against OA. Did the author mean that they investigated whether ERRγ could be a possible therapeutic target (and not molecule) to counteract/slow OA?

[Response]

It is a typographic error. We invested whether GSK5182 could be a possible therapeutic molecule against OA. We have revised this sentence (line 160).

Reviewer 3 Report

GSK5182, 4-hydroxytamofifen analog, is a new 3 potential therapeutic drug for osteoarthritis

By Min et al.,

The authors provide a novel approach in OA therapy adressing the Estrogen-related receptor gamma mediated pathway in cartilage. The topic is novel. The study hampers at several points. The discussion remains superficial as stated below.

The only semiquantitative data is OA scoring. Hence, data of western blot experiments should be evaluated semiquantitatively by densitimetric evaluation. The molecular weight of the detected bands is not provided and hence, not the information whether the activated form (cleaved) or the proenzyme has been visualized by blotting. In the method section some relevant information as indicated below is lacking.

Novel literature should be cited (e.g. at line 196„Reportedly, ERRs have functions in chondrocytes and OA.“:
Estrogen-Related Receptor γ Induces Angiogenesis and Extracellular Matrix Degradation of Temporomandibular Joint Osteoarthritis in Rats.

Zhao H, Liu S, Ma C, Ma S, Chen G, Yuan L, Chen L, Zhao H. Front Pharmacol. 2019 Nov 6;10:1290. doi: 10.3389/fphar.2019.01290. eCollection 2019.PMID: 31780931 

Lines 97-98: „overexpression of ERRg caused MMP-3 and MMP-13 expression at the mRNA and protein levels“ how about the proinfl. cytokine expression in response to overexpression ? – is it (increase in MMPs) possibly an indirect effect of cytokines? See also the conclusion in the discussion (lines 204-206) which might not be justified: „ERRg overexpression via infection with Ad-Esrrg dramatically elevated the mRNA or protein levels of MMP-3 and MMP-13 (Fig. 2). These results suggest that, in articular chondrocytes, pro-inflammatory cytokines and ERRg are closely associated.“

„three independent experiments“ (see Fig. Legends). Where these experiments performed with cells from different donors?

Abstract

Line 28: „in mouse articular culture chondrocytes.“ Better: „in mouse articular chondrocytes culture.“

Introduction

Line 65-66: the last sentences stands alone without support by a citation

Results

Figure 1: western blot: label the proteins with the molecular weight calculated from the electrophoresis. Legend (line 82) the „and“ should be shifted after (D)

„as internal markers“ write „reference“ see also later (Fig. 2).

Line 95: correct English grammar and style

Line 126: „sclerosis“, add „subchondral“

Line 163: „(1ng ml-1 )“ insert a blank „1 ng“

Discussion

Line186-187: „IL-1β, TNF-a, and IL-6 appear to be the central pro-inflammatory cytokines involved in OA pathophysiology.“ Add references.

Until line 202: the first and second paragraph of discussion does not discuss any result (rather introduction).

Line 204: „via infection with Ad-Esrrg“ write „transduction“

Line 203-217: the results are more or less recapitalated but not discussed with recent literature (there is no citation…)

Method section

The sentence sounds curious: Line 230: „Mice were harvested eight weeks after DMM“

Better: „samples/specimens were harvested“

4.3

Line 234: age of the mice? 235: „The cells were maintained as a monolayer“ how long, passage of culturing?

Line 238: chondrocyte cell number? „Mouse articular chondrocytes were cultured for two days, infected with various concentrations of Ad-Esrrg adenoviruses“ „cells were treated with various inflammatory cytokines (IL-1b, IL-6, and TNF-a) for 24 h“ please list the concentration and the sources of the cytokines

Line 243: how was concentration and quality of RNA checked?

4.5 western blotting: was a densitometric evaluation performed

Author Response

Reviewer #3:

The authors provide a novel approach in OA therapy adressing the Estrogen-related receptor gamma mediated pathway in cartilage. The topic is novel. The study hampers at several points. The discussion remains superficial as stated below.

Comment 1.

The only semiquantitative data is OA scoring.

[Response]

We agreed with the reviewer's comment. To overcome this limitation, we used a sufficient mouse number, e.g., 10 mice for IA injection, 25 mice for Tg mice DMM surgery, and 16 mice for the knockout mice DMM surgery.

Comment 2.

Hence, data of western blot experiments should be evaluated semiquantitatively by densitimetric evaluation. The molecular weight of the detected bands is not provided and hence, not the information whether the activated form (cleaved) or the proenzyme has been visualized by blotting.

[Response]

According to the reviewer's comment, we provided the semiquantitative densitometry evaluation data in all western blot experiments. Also, we have provided all molecular weight information in each Figure and uncropped blot images in Supplementary figures.   

Comment 3.

In the method section, some relevant information as indicated below is lacking.

[Response]

N/A

Comment 4.

Novel literature should be cited (e.g. at line 196„Reportedly, ERRs have functions in chondrocytes and OA.“:

Estrogen-Related Receptor γ Induces Angiogenesis and Extracellular Matrix Degradation of Temporomandibular Joint Osteoarthritis in Rats.

Zhao H, Liu S, Ma C, Ma S, Chen G, Yuan L, Chen L, Zhao H. Front Pharmacol. 2019 Nov 6;10:1290. doi: 10.3389/fphar.2019.01290. eCollection 2019.PMID: 31780931

[Response]

We cited this novel literature (line 216, line 224).

Comment 4.

Lines 97-98: „overexpression of ERRg caused MMP-3 and MMP-13 expression at the mRNA and protein levels“ how about the proinfl. cytokine expression in response to overexpression ? – is it (increase in MMPs) possibly an indirect effect of cytokines? See also the conclusion in the discussion (lines 204-206) which might not be justified: „ERRg overexpression via infection with Ad-Esrrg dramatically elevated the mRNA or protein levels of MMP-3 and MMP-13 (Fig. 2). These results suggest that, in articular chondrocytes, pro-inflammatory cytokines and ERRg are closely associated.“

[Response]

Based on our previous report and other reports, ERRg might directly regulate the MMPs transcriptions, not through indirect effects of cytokines (Nature communications 2017, 8, 2133; Front Pharmacol 2019, 10, 1290). We have discussed this issue in the discussion section (lines 214~216)

Comment 5.

„three independent experiments“ (see Fig. Legends). Where these experiments performed with cells from different donors?

[Response]

We used chondrocytes from different donors (pups). We have added this information in Fig 1&2. Legends (lines 88~89, lines 117~118)  

Abstract

Comment 6.

Line 28: „in mouse articular culture chondrocytes.“ Better: „in mouse articular chondrocytes culture.“

[Response]

We edited this sentence (line 28)

Introduction

Comment 7.

Line 65-66: the last sentences stands alone without support by a citation

[Response]

We added references in this sentence (line 68)

Results

Comment 8.

Figure 1: western blot: label the proteins with the molecular weight calculated from the electrophoresis.

[Response]

Following the reviewer’s comment, we labeled each protein's molecular weight in Figure with uncropped blot images in supplementary figures.

Comment 9.

Legend (line 82) the „and“ should be shifted after (D)

[Response]

We relocated this word (line 86)

Comment 10.

„as internal markers“ write „reference“ see also later (Fig. 2).

[Response]

The GAPDH, b-actin, and lamin B are well known internal markers. It is not necessary to cite the references.

Comment 11.

Line 95: correct English grammar and style

[Response]

We edited these sentences (line 102)

Comment 12.

Line 126: „sclerosis“, add „subchondral“

[Response]

We added “subchondral” to these sentences (line 136)

Comment 13.

Line 163: „(1ng ml-1 )“ insert a blank „1 ng“

[Response]

We edited it (line 172)

Discussion

Comment 14.

Line186-187: „IL-1β, TNF-a, and IL-6 appear to be the central pro-inflammatory cytokines involved in OA pathophysiology.“ Add references.

[Response]

We added appropriate references (line 201)

Comment 15.

Until line 202: the first and second paragraph of discussion does not discuss any result (rather introduction).

[Response]

We re-arranged the discussion section with a discussion of results (lines 192~198, 205~207, 214~216, 233~236)

Comment 16.

Line 204: „via infection with Ad-Esrrg“ write „transduction“

[Response]

We edited it (line 211)

Comment 17.

Line 203-217: the results are more or less recapitulated but not discussed with recent literature (there is no citation…)

[Response]

We re-arranged the discussion section by adding recent literature and discussion of results (lines 191~198, 204~206, 212~215, 232~234)

Method section

Comment 18.

The sentence sounds curious: Line 230: „Mice were harvested eight weeks after DMM“

Better: „samples/specimens were harvested“

[Response]

We edited according to the comment (line 252)

Comment 18.

4.3 Line 234: age of the mice? 235: „The cells were maintained as a monolayer“ how long, passage of culturing?

[Response]

We used 4 days old pubs for the chondrocyte culture, and we used only "0" passage primary chondrocytes in all experiments. We added this information in the text (line 257~258)

Comment 19.

Line 238: chondrocyte cell number? „Mouse articular chondrocytes were cultured for two days, infected with various concentrations of Ad-Esrrg adenoviruses“ „cells were treated with various inflammatory cytokines (IL-1b, IL-6, and TNF-a) for 24 h“ please list the concentration and the sources of the cytokines

[Response]

Following the reviewer’s comment, we have added chondrocyte cell number, Ad-Esrrg concentration, and inflammatory cytokines concentrations (lines 257~267).

Comment 20.

Line 243: how was concentration and quality of RNA checked?

[Response]

We checked the quality and concentrations of RNA with nanodrop equipment. We have added this information to the text (lines 269~270).

Comment 21.

4.5 western blotting: was a densitometric evaluation performed

[Response]

Following the reviewer’s comment, we have performed a densitometric evaluation in all western blotting.

Reviewer 4 Report

The paper of Dr. Son is not original. It almost completely repeats the results of his recent paper (Nature Communs 2017, 8:2133). However, the present version is written and arranged in a worst possible way both related to language use and data presentation.

Author Response

Reviewer #4:

The paper of Dr. Son is not original. It almost completely repeats the results of his recent paper (Nature Communs 2017, 8:2133). However, the present version is written and arranged in a worst possible way both related to language use and data presentation.

[Response]

Our previous study reported that ERRγ acts as a catabolic regulator of cartilage degeneration and OA pathogenesis (Nature Communs 2017, 8:2133). We have also demonstrated that the inverse agonist of ERRγ, GSK5182, inhibits OA pathogenesis in a mouse model. However, previous studies have reported limited information regarding the relationship between pro-inflammatory cytokines, ERRγ expression, and GSK5182. Therefore, we further investigated the relationship between pro-inflammatory cytokines and ERR γ expression, and the GSK5182 function in the pro-inflammatory cytokine-mediated cartilage catabolism in the OA joint. The novel approach of this study is that we examined whether the ERRγ inverse agonist GSK5182 blocks pro-inflammation mediated MMP-3 and MMP-13 expression. Our results demonstrated that treatment with GSK5182 dramatically inhibited IL-1b, IL-6-, or TNF-a induced MMP-3 and MMP-13 expression in primary cultured chondrocytes (Fig. 4A, B, C). Since pro-inflammatory cytokines are primary targets for osteoarthritis, only a few clinical studies have investigated. This study will promote the development of a therapeutic drug for OA by blocking inflammatory pathways, and we suggest that GSK5182 is a potential candidate molecule.

We have discussed the differences and novelty in this manuscript compared with our previous paper (lines 57~58, 68~69, 192-198, 229~238).

Round 2

Reviewer 1 Report

The response to the previous comments is partial.

Abstract: describe the advancement in this study from the previous study.

Authors should cite recent reviews/articles in inflammation for example PMID: 21788902, 32768946, 27389927, 28915300. Authors should also provide details for why the hydroxy tamoxifen analog is important to study. What is the difference in the compound used in this study and 4-hydroxy tamoxifen? Also discuss the limitation of long term ERR inhibition on cartilage and bone health. Many previous studies have shown that inhibition of ERR with tamoxifen induces bone formation, e.g. PMID: 15576459. Discuss the findings in this study (reduced bone remodeling) upon ERR inhibition in the light of published research work.  

which primer was used for qPCR? Was it the same as used for PCR in Fig 1A. 1E etc? Which method was used for qPCR (TaqMan or SYBR green)? Authors should provide the sequence for qPCR primers.

Figure 3: Authors should plot the OARSI score for the sham surgery and use appropriate statistical method (depending on data distribution, parametric or non-parametric) for comparison as t-test or Mann-Whitney is not the suitable test for comparing more than two groups. Also describe the method used to analyze data distribution. I suggest incorporating the effect size and power of the analysis at least for the in vivo experiments.

Figure 4E: did the authors find any difference in bone remodeling. In the graph, reduce the height of y-axis.

Validation of ERR overexpression in adenovirus experiment will improve the scientific rigor.

Highlight the areas of bone sclerosis and osteophytes in the figure.

Authors have used both, sagittal and coronal sections of knee. This should be mentioned for the images where coronal or sagittal sections are used and should describe if there is advantage of one method over the other.

Justify the use of only male mice in this study.

Author Response

Comment 1.

Authors should cite recent reviews/articles in inflammation for example PMID: 21788902, 32768946, 27389927, 28915300.

[Response]

  We have included a detailed discussion of these references and cited them throughout the introduction section.

Comment 2.

Authors should also provide details for why the hydroxy tamoxifen analog is important to study. What is the difference in the compound used in this study and 4-hydroxy tamoxifen?

[Response]

Following the reviewer’s comment, we have added a discussion of the differences between 4-hydroxytamoxifen (4-OHT) and 4-OHT analog (GSK5182) in the Discussion section (lines 237–242).

“The ERRγ binding activity for synthetic ligands diethylstilbestrol (DES), tamoxifen (TAM), and 4-hydroxytamoxifen (4-OHT) was evaluated and 4-OHT was shown to be the most specific and had high binding affinity with ERRγ at the micro-molar level [46]. Synthetic GSK5182, a 4-OHT analog, was developed to exhibit higher affinity (IC50 = 79 nM) for ERRγ [47] and was shown to regulate the transcriptional activity of ERRγ [48,49].” 

Comment 3.

Also discuss the limitation of long term ERR inhibition on cartilage and bone health. Many previous studies have shown that inhibition of ERR with tamoxifen induces bone formation, e.g. PMID: 15576459. Discuss the findings in this study (reduced bone remodeling) upon ERR inhibition in the light of published research work.

[Response]        

At the reviewer’s suggestion we have included a discussion of this in the revised manuscript (lines 248–251).

“Treatment with tamoxifen induced cancellous bone and longitudinal growth, while treatment with GSK5182 reduced DMM-induced bone remodeling including cartilage destruction, osteophyte development, and subchondral bone sclerosis [7,51].”

Comment 4.

which primer was used for qPCR? Was it the same as used for PCR in Fig 1A. 1E etc? Which method was used for qPCR (TaqMan or SYBR green)? Authors should provide the sequence for qPCR primers.

[Response]        

We used the same primers for both the PCR and qPCR assays. The qPCR experiments were completed using SYBR premix ExTaq reagents (TaKaRa Bio), as described in the Materials and Methods section (lines 287-288).

Comment 5.

Figure 3: Authors should plot the OARSI score for the sham surgery and use appropriate statistical method (depending on data distribution, parametric or non-parametric) for comparison as t-test or Mann-Whitney is not the suitable test for comparing more than two groups. Also describe the method used to analyze data distribution. I suggest incorporating the effect size and power of the analysis at least for the in vivo experiments.

[Response]        

We replaced the OARSI score with a scatter plot for all the animal experiments. We used a two-tailed Student’s t-test for the pairwise comparisons of the in vitro data (none vs treatments), and a non-parametric Mann-Whitney U test for the in vivo data (Ad-C vs. Ad-Esrrg, WT DMM vs. TG or KO DMM). Data distribution was evaluated for normalcy using the Shapiro-Wilk test. We have described this statistical analysis in detail in the statistical analysis section (lines 311–317). We also added a description of the U-value, p-value, n number, and effect size (r) to the Figure 3 and 4 legends.

Comment 6.

Figure 4E: did the authors find any difference in bone remodeling. In the graph, reduce the height of y-axis.

[Response]        

We only checked for cartilage degeneration in this experiment (completed at 3 weeks). Based on our previous experiments, bone remodeling might be reduced over a longer evaluation period (8 weeks).

Comment 7.

Validation of ERR overexpression in adenovirus experiment will improve the scientific rigor.

[Response]        

At the reviewer’s suggestion we have added the ERRγ expression data from IHC to Figure 4F.

Comment 8.

Highlight the areas of bone sclerosis and osteophytes in the figure.

[Response]        

We performed 3 weeks experiment with lateral section. Therefore, we could not see the bone sclerosis and osteophytes in the figure

Comment 9.

Authors have used both, sagittal and coronal sections of knee. This should be mentioned for the images where coronal or sagittal sections are used and should describe if there is advantage of one method over the other.

[Response]        

We used sagittal (lateral) sections for the 3 week evaluations (Fig. 3A and Fig. 4E) or coronal (frontal) sections for the 8 week evaluations (Fig. 3B and C). We were able to evaluate both cartilage degeneration and synovitis in the sagittal (lateral) sections from the 3 week evaluations, while the evaluations of bone remodeling including cartilage destruction, osteophyte development, and subchondral bone sclerosis were more easily evaluated using coronal (frontal) sections, thus we made a change for the 8-week DMM experiments. We have described each method in the relevant figure legends. 

Comment 10.

Justify the use of only male mice in this study.

[Response]        

To avoid developmental differences in the OA experiments, where these differences may be associated with hormonal effects, we elected to use only 12-week-old male mice for the DMM surgery or IA injection treatments (lines 264-265)

Reviewer 2 Report

 A better manuscript after the revision.

I only noticed that the authors performed one way anova to analyze the protein expression. However, they forgot to update the statistical analysis paragraph of the methods. 

Author Response

Comment 1.

I only noticed that the authors performed one way anova to analyze the protein expression. However, they forgot to update the statistical analysis paragraph of the methods. 

[Response]        

We have updated this information in the statistical analysis section (lines 311~317)

Reviewer 3 Report

The manuscript has substantially been revised. Additional information was provided. Semiquantitative analysis (densitometry) of western blotting was provided. Some minor aspects/inconsistencies could be changed during proof reading such as:

line 45: insert a blank

line 58: abbreviate "osteoarthritis": "OA" as before

"clinical studies...investigated" better: "clinical studies...have been performed"

line 102: extracellularly

"quantification" should be substituted by "semiquantification" since no absolute amount of protein can be concluded

line 265: "PFU" not sure whether this abbreviation was explained

Author Response

The manuscript has substantially been revised. Additional information was provided. Semiquantitative analysis (densitometry) of western blotting was provided. Some minor aspects/inconsistencies could be changed during proof reading such as:

Comment 1.

line 45: insert a blank

[Response]        

We have inserted a blank

Comment 2.

line 58: abbreviate "osteoarthritis": "OA" as before

[Response]        

We have reintroduced our osteoarthritis abbreviation “OA”.

Comment 3.

"clinical studies...investigated" better: "clinical studies...have been performed"

[Response]        

We have edited this sentence

Comment 4.

line 102: extracellularly

[Response]        

We have replaced this word

Comment 5.

"quantification" should be substituted by "semiquantification" since no absolute amount of protein can be concluded

[Response]        

We have made this change in the relevant figure legends.

Comment 6.

line 265: "PFU" not sure whether this abbreviation was explained

[Response]        

We have added an explanation of PFU to the relevant figure legends.

Reviewer 4 Report

[Response]

Our previous study reported that ERRγ acts as a catabolic regulator of cartilage degeneration and OA pathogenesis (Nature Communs 2017, 8:2133). We have also demonstrated that the inverse agonist of ERRγ, GSK5182, inhibits OA pathogenesis in a mouse model. However, previous studies have reported limited information regarding the relationship between pro-inflammatory cytokines, ERRγ expression, and GSK5182. Therefore, we further investigated the relationship between pro-inflammatory cytokines and ERR γ expression, and the GSK5182 function in the pro-inflammatory cytokine-mediated cartilage catabolism in the OA joint. The novel approach of this study is that we examined whether the ERRγ inverse agonist GSK5182 blocks pro-inflammation mediated MMP-3 and MMP-13 expression. Our results demonstrated that treatment with GSK5182 dramatically inhibited IL-1b, IL-6-, or TNF-a induced MMP-3 and MMP-13 expression in primary cultured chondrocytes (Fig. 4A, B, C). Since pro-inflammatory cytokines are primary targets for osteoarthritis, only a few clinical studies have investigated. This study will promote the development of a therapeutic drug for OA by blocking inflammatory pathways, and we suggest that GSK5182 is a potential candidate molecule.

We have discussed the differences and novelty in this manuscript compared with our previous paper (lines 57~58, 68~69, 192-198, 229~238).

Review for the revised version

The authors response is not honest as:

  1. Fig1b repeats the data presented in Fig2e of the recent paper (Nature Communs 2017, 8:2133).
  2. Fig 1d contains data of Fig2f(left panel) of the recent paper (Nature Communs 2017, 8:2133).
  3. Fig4b is a complete copy of Fig7e of the recent paper (Nature Communs 2017, 8:2133).
  4. etc

The authors should remove ALL the duplicated data obtained from their previous study (Nature Communs 2017, 8:2133).

Only after all the required correction this paper could be reviewed.

Author Response

Comment 1.

  1. Fig1b repeats the data presented in Fig2e of the recent paper (Nature Communs 2017, 8:2133).

[Response]        

We analyzed the mRNA expression of ERRg in three kinds of pro-inflammatory treated chondrocytes. We added a description of the induction time points for ERRg and compared the ERRg mRNA expression for each treatment group.

Comment 2.

  1. Fig 1d contains data of Fig2f(left panel) of the recent paper (Nature Communs 2017, 8:2133).

[Response]        

We performed a comparison of ERRg mRNA expression among three kinds of pro-inflammatory cytokines.

Comment 3.

  1. Fig4b is a complete copy of Fig7e of the recent paper (Nature Communs 2017, 8:2133).

[Response]        

This experiment describes the inhibitory effects of GSK5182 on pro-inflammatory cytokines. We tested whether GSK5182 inhibits additional cytokines (IL-1b and TNF-a).

Round 3

Reviewer 1 Report

The authors have addressed all the previous comments. I congratulate the authors for the success of their work.

Reviewer 4 Report

The author"s responses are not satisfactory.

The authors were supposed to remove ALL the duplicated data obtained from their previous study (Nature Communs 2017, 8:2133). However, the authors did not make any changes in accordance with the Comments for version R1 (see below).

The authors either should remove the duplicated data from the manuscript or to request a permission from Nature Communs for reproducing previously published results and to indicate in the present manuscript which results are republished if the permission from Nature Communs is obtained.

Author's Notes

Comment 1.

  1. Fig1b repeats the data presented in Fig2e of the recent paper (Nature Communs 2017, 8:2133).

[Response]        

We analyzed the mRNA expression of ERRg in three kinds of pro-inflammatory treated chondrocytes. We added a description of the induction time points for ERRg and compared the ERRg mRNA expression for each treatment group.

Comment 2.

  1. Fig 1d contains data of Fig2f(left panel) of the recent paper (Nature Communs 2017, 8:2133).

[Response]        

We performed a comparison of ERRg mRNA expression among three kinds of pro-inflammatory cytokines.

Comment 3.

  1. Fig4b is a complete copy of Fig7e of the recent paper (Nature Communs 2017, 8:2133).

[Response]        

This experiment describes the inhibitory effects of GSK5182 on pro-inflammatory cytokines. We tested whether GSK5182 inhibits additional cytokines (IL-1b and TNF-a).